# Does increasing social presence enhance the effectiveness of writing explanations?

**Leonie Jacob**[1]*, **Andreas Lachner**[2], **Katharina Scheiter**[1,2]

**1** Leibniz-Institut für Wissensmedien, Tübingen, Germany, **2** University of Tübingen, Tübingen, Germany

* l.jacob@iwm-tuebingen.de

**Data Availability Statement:** All relevant data are within the manuscript and its Supporting information files.

**Funding:** The research reported in this article was supported by the Federal Ministry of Education and

## Abstract

Writing explanations has demonstrated to be less effective than providing oral explanations, as writing triggers less amounts of perceived social presence during explaining. In this study, we investigated whether increasing social presence during writing explanations would aid learning. University students ($N = 137$) read an instructional text about immunology; their subsequent task depended on experimental condition. Students either explained the contents to a fictitious peer orally, wrote their explanations in a text editor, or wrote them in a messenger chat, which was assumed to induce higher levels of social presence. A control group retrieved the material. Surprisingly, we did not obtain any differences in learning outcomes between experimental conditions. Interestingly, explaining was more effortful, enjoyable, and interesting than retrieving. This study shows that solely inducing social presence does not improve learning from writing explanations. More importantly, the findings underscore the importance of cognitive and motivational conditions during learning activities.

## Introduction

Generating explanations is regarded as a successful strategy to enhance students' understanding, as it triggers generative processes associated with deep learning [1–8]. Seminal studies on learning by explaining started to investigate the role of explaining in interactive settings, such as during collaborative learning or tutoring, in which the explainer received feedback from the recipient, for instance, in form of direct questions [6,8]. Results showed that students who explained learned material were engaged in deeper learning processes and showed higher learning outcomes compared to restudying [5–8]. Recent research replicated the beneficial of explaining in non-interactive settings, in which no recipient was present; thus, students explained learned material to a *fictitious* peer which also resulted in higher learning outcomes [1,9,10]. Moreover, learning by explaining was more often shown to be effective when students were required to generate oral explanations instead of written ones [11–13]. A possible reason for the benefit of oral explaining might be the difference between perceived *social presence* during explaining. In this context, Jacob, Lachner, and Scheiter [13] provided first evidence that writing explanations induces lower levels of social presence during explaining than providing oral explanations, and reduces the quality of the generated explanations [11–13]. Social

Research in Germany (BMBF) in the form of a contract awarded to AL (01JA1611).

**Competing interests:** The authors have declared that no competing interests exist.

presence is a central concept in discourse theory, and is commonly defined as the extent to which a person feels that a communication partner is present during a mediated conversation, such as in online learning environments [14–18]. In this context, social presence not only holds true for real persons, but also for virtual or even fictitious communication partners [19–21]. Given that writing explanations is a learning activity that can easily be implemented in learning contexts, the question arises whether and how writing explanations can be made more effective. As previous studies highlighted the role of social presence, we investigated whether inducing social presence would increase the effectiveness of writing explanations regarding students' comprehension. Additionally, as recent studies showed that learning by explaining affected (meta-)cognitive and motivational factors [22,23], we explored whether perceived mental effort and subjective difficulty as central facets of perceived cognitive load [24,25], and students' monitoring accuracy as a metacognitive factor [3] differed across experimental conditions. Further, we investigated whether students' enjoyment and interest as crucial facets of students' valence-related motivational orientations [26] varied among conditions.

## Learning by explaining to fictitious peers

Learning by explaining is a generative learning activity which aims at enhancing students' meaningful learning [27]. In line with the generative learning theory, the process of explaining as a generative act may elicit cognitive (e.g., mental effort) and metacognitive (e.g., monitoring) processes, which should contribute to students' comprehension: First, and in line with Mayer's SOI model [28], when explaining students need to *select* the most relevant information if the provided materials and to *organize* the information in a coherent way. Then, they need to connect the new contents with their already existing knowledge to *integrate* them into their long-term memory [27,29,30]. Through this connection, students are able to provide explanations that include further details and information that go beyond the giving materials, which results in new knowledge and meaningful learning [27–34]. This process additionally triggers students to monitor whether they understood all relevant contents correctly or whether they need to restudy specific information. As a consequence, students' metacognitive monitoring may become more accurate when explaining, which has also been observed for other generative learning activities such as keyword generation or gap filling [3,4,35,36]. Learning by explaining is commonly implemented in interactive learning settings in which students explain learned contents to present and interactive peers; this setting allows students to exchange ideas and thought, which additionally enhances their understanding [6,7,31,37–47]. Interestingly, recent research started to investigate the effectiveness of explaining to a *fictitious* peer and reported promising results [1,2,4,12,13,23]. For instance, Fiorella and Mayer [1] conducted an experiment in which university students first studied a text with the expectation to either answer a test (test expectancy) or to explain the content to a fictitious peer (explaining expectancy) after a learning phase. Then, they were randomly assigned to the experimental conditions: They either restudied the material (no explaining condition) or explained the contents to a fictitious peer by generating a video (explaining condition). Results demonstrated that explaining expectancy had an effect only on students' short-term retention ($d = 0.55$, medium effect), whereas the actual act of explaining learned material resulted in better performance regarding their long-term retention ($d = 0.56$, medium effect, see also 22, for replications).

In another study, Hoogerheide, Visee, Lachner, and van Gog [23] demonstrated that learning by explaining did not only result in higher learning outcomes but also affected students' motivation during learning. The authors conducted a study with three conditions: Students either explained learned material to a fictitious peer (explaining condition) or wrote a

summary about the contents (summarizing condition). A control group restudied the materials (restudy condition). Results showed that students in the explaining condition outperformed students in the restudy condition. Interestingly, students' enjoyment mediated the explaining effect, as students enjoyed explaining more than summarizing, which, in turn, yielded better comprehension. Additionally, students who explained the materials reported higher levels of invested mental effort during the learning task compared to restudying, which, however, was not linked to higher learning outcomes [23].

Furthermore, several studies indicated that generating explanations additionally supports students' monitoring accuracy, which is a crucial metacognitive facet of successful learning [3,4]. For instance, during reorganizing and connecting learned contents for generating an explanation, students might detect unsolved problems, misunderstandings, or missing information that prevent them from understanding the contents deeply [27]. This detection helps students to judge their current understanding more accurately and supports them in restudying information which they did not understand clearly to reach their learning aims [48]. In a study by Fukaya [3], for instance, students first read five different texts with the intention to either explain the contents or to write keywords about the texts. A control group only had the intention to explain the content but did not actually generate an explanation. Results showed a difference among conditions: Students who actually generated an explanation judged their comprehension more accurately than students who only had the intention to explain or who wrote keywords ($d = 0.91$, large effect). Thus, the actual act of explaining seems to increase students' monitoring accuracy.

Even though serval studies indicated beneficial effects of learning by explaining to fictitious others on students' comprehension and monitoring skills, little is yet known about the underlying mechanism of why explaining is effective. Recently, researchers emphasized the role of social presence during explaining [11–13,49], as higher levels of social presence (which also may arise from virtual of even fictitious characters) are linked to central components of learning, such as cognition or motivation [14,18–21,50,51]. On the one hand, the social presence of a fictitious communication partner may engage students to adapt their knowledge to the audience's needs [52,53], for instance, by providing further details and elaborations that go beyond the contents of the learning material. Such audience-adjustments may result in deeper elaborative processes and contribute to meaningful learning [54]. On the other hand, from a motivational perspective [23], the social presence of a fictitious person may also increase the feeling of relatedness [55]. Higher levels of relatedness may yield higher levels of enjoyment and investments during providing an explanation, and, in turn, contribute to comprehension [23,55,56].

## Learning by writing explanations

Although prior research documented beneficial effects of explaining compared to retrieving, recent studies demonstrated that generating oral explanations is more beneficial than writing an explanation. In an experiment by Hoogerheide, Deijkers, Loyens, Heijltjes, and van Gog [12] university students first read an instructional text about syllogistic reasoning, and then either wrote an explanation to a fictitious peer or restudied the learning material. In contrast to prior research on oral explaining, results indicated that writing an explanation did not result in higher learning outcomes than restudying. Therefore, the authors directly compared the influence of the explanatory modality in a second experiment. Results showed that only students who explained orally ($d = 0.43$, medium effect), but not in written form ($d = 0.19$, small effect) outperformed students who restudied the learning material. However, there was no significant difference between oral and written explanations regarding students' learning

outcome. Additionally, similar to Hoogerheide, Visee, Lachner, and van Gog [23], results indicated that students who generated an explanation in oral ($d$ = 1.96, large effect) or in written form ($d$ = 0.93, large effect) invested higher levels of mental effort during the learning task compared to students who restudied the material. Relatedly, Lachner, Ly, and Nückles [11] provided students with a Wikipedia article about combustion engines; after studying it, students were asked to explain the contents to a fictitious peer in either oral or written form. The findings revealed that students who generated an oral explanation reached higher scores in the comprehension posttest compared to students who wrote an explanation ($d$ = 0.67, medium effect), which could be explained by more elaborated explanations given in the oral condition. The authors attributed the findings to the fact that they used more difficult learning materials than Hoogerheide, Deijkers, Loyens, Heijltjes, and van Gog [12]. Against this background, Jacob, Lachner, and Scheiter [13] conducted a further experiment to resolve the conflicting findings regarding the explanatory modality. Results revealed an interaction effect between learning activity and text difficulty ($d$ = 0.31). The effect of explaining was only significant in the high-difficult condition, but not in the low-difficult condition. Thus, the explaining effect only held true when students learned from difficult but not from less difficult material. More interestingly, students who explained the contents to a fictitious peer orally outperformed students who wrote an explanation. Again, this effect was only significant when the learning material was difficult, but not when it was less complex. Additionally, perceived social presence and the richness of explanations mediated the explanatory effect: Students who explained orally perceived a stronger presence of the fictitious peer (measured by the number of personal references) compared to students who wrote an explanation. Higher levels of social presence, in turn, was associated with richer explanations (measured by the number of mentioned concepts) which resulted in better learning outcomes, at least for the difficult text. Apparently, the social presence during explaining accounted for the superiority of oral explaining. Furthermore, students who explained orally judged their current understanding more precisely than students who wrote an explanation and invested more mental effort than students who retrieved the materials [13,23].

These findings are in line with literature in applied linguistics, in which it is generally argued that writing is a rather solitary process, since the writer is normally separated from the audience regarding time and place [57,58]. Due to the lack of a social presence during written activities, the writer in contrast tends to adopt a knowledge-telling perspective, and as such only retrieves the content without adapting the content to a particular audience's needs [59]. These detrimental effects may increase, as writing often induces higher levels of cognitive load than speaking, which can be considered as an automated and less demanding process compared to writing. Overall, the lower levels of audience adjustments may be less conducive to learning [52], as, for instance, students provide lower amounts of examples to elaborate the content.

## The present study: Inducing social presence to enhance writing explanations

We conducted an experiment to investigate whether learning by writing explanations could be supported by inducing social presence during explaining. We used validated experimental materials [13,60], and provided university students with a text about immunology during the study phase. Afterwards, they were randomly assigned to one of four conditions: They explained the contents in oral form (oral condition), wrote their explanations in a text editor (standard written condition), or wrote them in a chat messenger program (chat condition, see Fig 1). In this messenger program, students could see a profile picture and a message from the

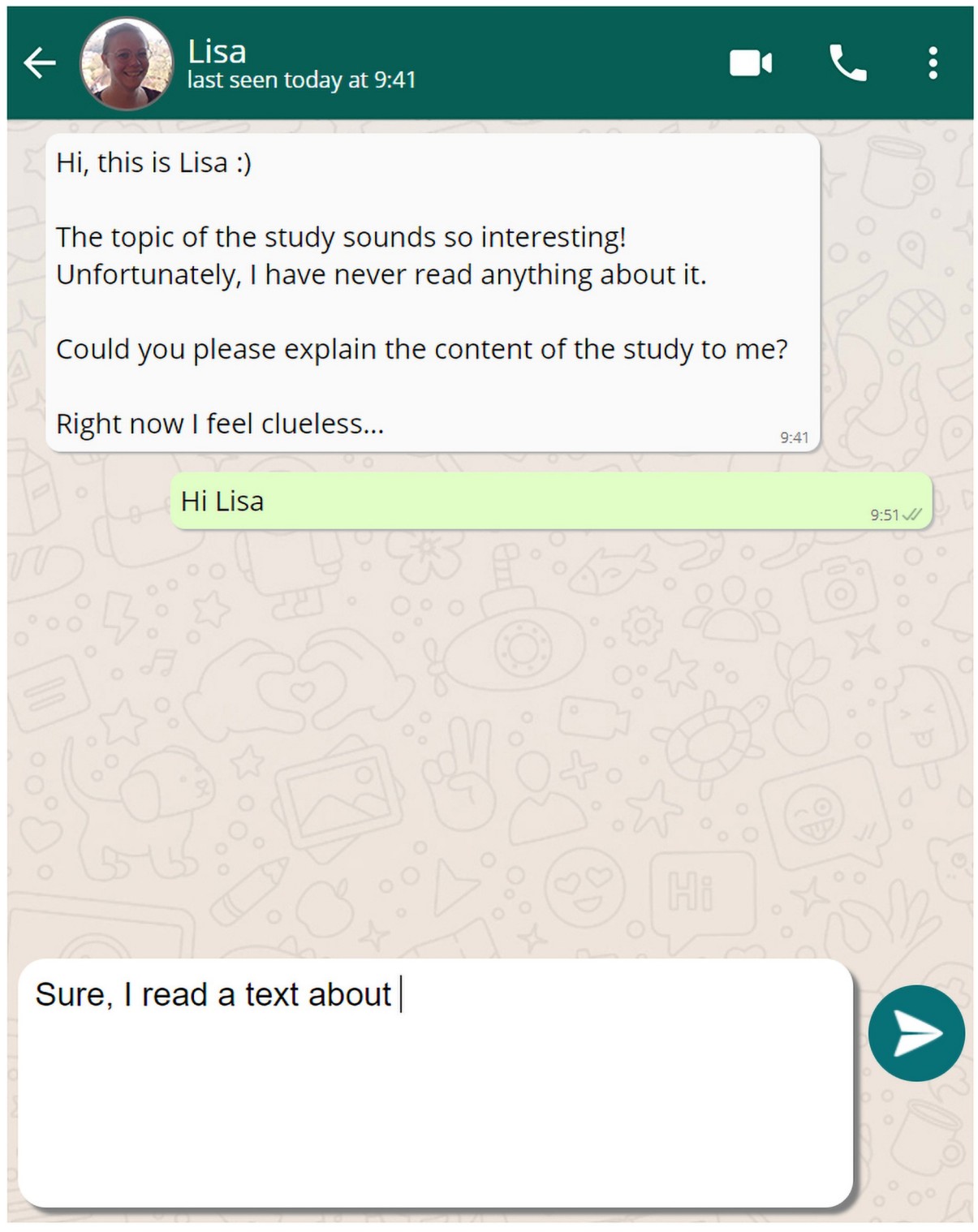

**Fig 1. Simulated mockup messenger chat for chat condition with induced social presence.** Mockup messenger chat with a profile picture and message from the fictitious student Lisa. Students in the chat condition could send text messages which appeared in the chat afterwards. We received written approval to publish the picture of the corresponding individual who completed the consent form for publication in a PLOS journal.

fictitious peer, which we assumed would induce higher levels of social presence [18]. Students in the control condition were asked to retrieve the contents (retrieval condition). We stated the following hypotheses, which were preregistered on AsPredicted.org (https://aspredicted. org/3nu5m.pdf).

In line with previous evidence, we hypothesized that students who generate an explanation (oral or written form) outperform students who retrieve the material regarding students' comprehension posttests (*Hypothesis 1*). Additionally, based on previous findings, we hypothesized that students who explain orally outperform students who write an explanation (standard written condition) to a fictitious peer (*Hypothesis 2*). Furthermore, we hypothesized that students who write an explanation with increased social presence (chat condition) outperform students in the standard written condition (*Hypothesis 3*). Additionally, we explored potential differences between the oral condition and the chat condition and assumed comparable outcomes regarding students' comprehension since social presence was induced in the chat condition aiming at reaching a similar level of perceived social presence as explaining orally.

Based on previous research, we additionally investigated potential differences regarding (meta-)cognitive (i.e., monitoring accuracy, mental effort, and subjective difficulty; 3, 13) and motivational factors, such as enjoyment and interest [22,23]. To account for the quality of the generated explanations, we measured three characteristics of the generated explanations (i.e., personal references, concepts, elaborations), which are commonly measured in research on learning by explaining [11–13].

## Method and methods

### Participants and design

The current study was approved in written form by the ethics committee of the Leibniz-Institut für Wissensmedien in Tübingen (approval number: LEK2019/009). We recruited university students ($N = 137$) from study programs that were not related to the study topic (i.e., biology). This sample reached the required sample size of 126 participants, as determined by an *a priori* power analysis. Power was set to.80, $\alpha$-error to.05, and the assumed effect size to $\eta_p^2 = .08$, as recent studies documented differences of medium to large effect size of explaining [11,12] and social presence in learning settings [14].

The mean age of the students was 23.23 years ($SD = 2.65$) and 73% of them were female. The students either stated to be German native speakers (85%), or that they grew up bilingually with German (15%). The students were advanced students, on average in their 5th semester ($SD = 3.19$) of their current study program and were mostly enrolled in humanities programs (71%). On average, they had taken biology classes in school for 8 years ($SD = 2.13$), had 10 points (on a scale from 0 to 15, corresponding to a B-) in their last report card in biology ($SD = 3.25$), and showed low to medium prior knowledge skills in the prior knowledge test ($M = 2.15$; $SD = 1.13$; on a scale from 0 to 5).

Students were randomly assigned to one of four experimental conditions (i.e., retrieval condition: $n = 34$; standard written condition: $n = 34$; chat condition: $n = 35$; oral condition: $n = 34$), which was the independent variable. The dependent variable was students' text comprehension, measured by two knowledge subtests comprising text-based questions and inference questions. To investigate potential (meta-)cognitive differences among conditions, we additionally collected data regarding students' perceived mental effort, subjective difficulty, and monitoring accuracy during the study phase (i.e., reading the learning material) as additional control variables, and during the learning activity (i.e., explaining vs. retrieving) as potential mediators. Moreover, to investigate potential differences regarding students' motivation during the learning activity, we asked them to rate their enjoyment and interest during

the learning activity. As potential underlying processes, we additionally analyzed three charac-
teristics of students' generated explanations: Personal references as an indicator for social pres-
ence, number of elaborations, and number of concepts to investigate further underlying
learning processes. As control variables, we measured students' prior knowledge, self-efficacy
in explaining, their biological interest at the beginning of the study, and their perceived social
presence during explaining.

## Study text

We used a validated text from the domain of biology from Golke and Wittwer [60]. The text
was about immunology and dealt with immune research based on laboratory mice. The text
was previously used in a study on learning by explaining [13]. Overall, the text had a length of
397 words and constituted a relatively complex text regarding common measures of text diffi-
culty [13].

## Prior knowledge test as control variable

We used the prior knowledge test from Golke and Wittwer [60], which contained five ques-
tions with an open-ended answer format and represented a multidimensional construct, mea-
suring different subcomponents of immunology (e.g., "Why can viruses be dangerous for
humans?"; McDonald's $\omega_t$ = .51). For each right answer students could receive one point,
yielding a maximum score of five points. Two independent raters coded 20% of the tests. As
interrater reliability was excellent ($ICC_{2,1}$ = .91), one rater coded the remaining answers [61].

## Knowledge posttests as dependent variables

We measured students' text-based knowledge and inference knowledge with two different
posttests from Golke and Wittwer [60]. The tests consisted of different questions than the
prior knowledge test and represented multidimensional constructs.

**Text-based questions.** The text-based questions comprised six open-ended questions,
which aimed at measuring students' basic knowledge of the text (e.g., "What is the main con-
cern against research with laboratory mice?"; McDonald's $\omega_t$ = .44). For each correct answer,
students could reach one point, resulting in a maximum score of six points. Two independent
raters coded 20% of the tests. As interrater reliability was excellent ($ICC_{2,1}$ = .95), one rater
coded the remaining answers.

**Inference questions.** The inference questions measured students' advanced understand-
ing (e.g., "How can the results in the text help to clarify the basic problems of immune research
with mice?"; McDonald's $\omega_t$ = .45), as students had to combine different aspects and informa-
tion across the text, and to draw conclusions. Again, students could receive one point per
answer, yielding a maximum score of six points. Two independent raters coded 20% of the
tests. As interrater reliability was excellent ($ICC_{2,1}$ = .92), one rater coded the remaining
answers.

## (Meta-)Cognitive processes during the learning activity

**Monitoring accuracy.** To measure students' monitoring accuracy, we asked them to
make prospective judgments about their expected performance on the posttest (i.e., text-based
and inference questions combined) to investigate their monitoring accuracy by rating the fol-
lowing item: "How confident are you that you can answer questions to the text correctly?"
[62,63]. Monitoring accuracy is commonly measured by the correspondence between students'
judgements of their own current understanding and their actual performance on a

comprehension test [60,64,65]. Therefore, students rated their comprehension after the study phase and after the learning activity on a scale from 0% (no confidence) to 100% (absolute confidence) [13,66]. We operationalized students' monitoring accuracy in terms of bias [4,13,60,67–69]. Bias refers to the signed difference between students' estimated performance and the actual performance (i.e., $X_{Judgment} - X_{Performance}$). Hence, this estimation indicated whether students over- or underestimate their own performance. Positive values indicate an overestimation and negative values indicate an underestimation of their judged performances. A value of zero indicates an accurate judgment.

**Cognitive load.**   We asked students to rate their perceived mental effort (i.e., "How much effort did you invest in explaining the material?"), as subjective proxies to perceived cognitive load, after the study phase and after the learning activity on a 9-point Likert scale from 1 "very little" to 9 "very much" [24].

Additionally, students rated their subjective difficulty (i.e., "How easy was it for you to explain the material?"), as subjective proxies to perceived cognitive load after the study phase and after the learning activity on a 9-point Likert scale from 1 "very easy" to 9 "very difficult" [25].

### Motivation during the learning activity

**Enjoyment during the learning activity.**   We measured students' enjoyment during the learning activity by using two self-generated items (e.g., "I enjoyed doing the task"). The students rated their enjoyment on a 4-point Likert scale from 1 "not at all" to 4 "absolutely". Reliability was excellent (McDonald's $\omega_t$ = .96).

**Interest in the learning activity.**   We measured students' enjoyment during the learning activity by using two self-generated items (e.g., "I enjoyed doing the task"). The students rated their interest on a 4-point Likert scale from 1 "not at all" to 4 "absolutely". Again, reliability was good (McDonald's $\omega_t$ = .81).

### Characteristics of explanations

Based on prior research, we analyzed three characteristics of the generated explanations as indictors for underlying processes during explaining: Personal references, concepts, and elaborations [11–13].

**Personal references.**   Based on discourse literature, we used personal references (i.e., "I", "you", etc.) as an indicator for the perceived social presence [57,58,70,71] which is frequently also used in learning by explaining literature [11–13]. We automatically counted the number of personal references with RStudio [72].

**Concepts.**   As an indicator of the level of comprehensiveness, we counted the number of concepts per explanation [73]. Concepts are the mentioned constructs within an explanation. For instance, the sentence "*Laboratory mice* grow up in sterile *environments* and, therefore, the *laboratory mice* do not have any *contact* with *diseases*" contains four concepts (marked in italics). Redundant concepts (e.g., "laboratory mice") were ignored [73]. Two independent raters coded 20% of the explanations. As interrater reliability was excellent ($ICC_{2,1}$ = .95), one rater coded the remaining explanations.

**Elaborations.**   As a third characteristic, we counted the number of elaborations within the explanations. An elaboration was operationalized as an idea unit, such as examples, analogies, and own experiences [2,11]. For instance, the sentences "Laboratory mice are too clean *because they have been cultivated over generations*" is an elaboration, as it was not mentioned in the study text. The student, therefore, combined the information of the text with her or his prior knowledge to generate the example. Two independent raters counted the number of

elaborations for 20% of all explanations. Interrater reliability was excellent, $ICC_{2,1}$ = .95. Therefore, one rater coded the remaining explanations.

## Additional control measures

**Perceived social presence.** We asked students to rate their feelings of a social presence form the fictitious peer during explaining. We asked students to rate three self-generated items (i.e., "How strongly did you imagine that Lisa was real?"; "How important was it for you that Lisa understands the contents?"; "How strongly did you perceived being in a communicative situation"; McDonald's $\omega_t$ = .84). Based on related approaches on subjective assessments of task characteristics, we used a 9-point Likert scale from 1 "not at all" to 9 "completely" [24,25]. Since students in the retrieval condition were not asked to explain to a fictitious peer, only students in explaining conditions (i.e., standard written condition, chat condition, oral condition) rated their perceived social presence.

**Self-efficacy in explaining.** We assessed students' self-efficacy in explaining as a further control variable. We used three adapted items (e.g., "I always find ways to explain even difficult contents"; McDonald's $\omega_t$ = .74) from Jerusalem and Schwarzer [74]. Students rated their explaining skills on a 4-point Likert scale from 1 "I completely disagree" to 4 "I completely agree".

**Biological interest.** We measured students' interest in biology as an additional control variable by using three items (e.g., "I am fascinated by biological topics"; McDonald's $\omega_t$ = .91), based on Kunter and colleges [75]. Students rated the items on a 4-point Likert scale from 1 "I completely disagree" to 4 "I completely agree".

## Procedure

The instructor welcomed the students and informed them about the study procedure. After providing written consent, all students were seated individually in front of a laptop in noise-cancelling cubicles so that they neither could see or hear each other. A maximum of 6 students could participate in one session. The entire study was self-paced and all instructions were provided in an online learning environment created by Klemke [76].

First, students rated their efficacy in explaining and their interest in biology. Then, they answered the prior knowledge test. Afterwards, they read the study text in a self-paced manner without the intent to explain or retrieve the material afterwards. After this study phase, they rated their mental effort and difficulty during reading, and provided a monitoring judgment. Then, the students were randomly assigned to one of four experimental conditions (i.e., retrieval condition, standard written condition, chat condition, oral condition). All students had 10 minutes to accomplish the learning activity. During the learning activity, they could take notes on a separate sheet but were not able to see the learning materials anymore. Students in the oral and standard written condition were given the following instruction:

> *Imagine the following scenario: One person could not participate in the study. However, she is highly interested in the topic of the study. She has not read anything about the topic yet. Therefore, she asks you to explain the central contents of the text. Please provide her a clear and detailed explanation, so that she can understand the content without additional information. You may take notes on a separate sheet.*

Students in the oral explanations recorded their explanation with a microphone which was connected to the laptop. Students in the standard written condition wrote their explanation

into a text editor box. In contrast, students in the chat condition received a more personalized instruction:

> *Here you can see a chat with Lisa. Lisa is highly interested in the topic of the study. She has not read anything about the topic yet. Therefore, she asks you to explain the central contents of the text. Please provide her a clear and detailed explanation, so that she can understand the content without additional information. You may take notes on a separate sheet.*

Students in the chat condition provided their explanation in a messenger mockup (Fig 1), which was an imitation of WhatsApp Messenger (freeware created by WhatsApp Inc. in 2009 and acquired by Facebook in 2014). We provided different social cues to induce higher levels of social presence (see Weidlich & Bastiaens for a similar approach; 18). First, students could see a profile picture from *Lisa*. Second, they received a short message from Lisa who directly asked the students for an explanation (Fig 1). We automatically adapted the time of the received message to increase the synchrony of the communication. Students could send Lisa a message to share their explanation within the chat. Students in the retrieval condition only retrieved the material via an open recall task [4,13] with the following instruction:

> *Please retrieve the information of the text. You may take notes on a separate sheet.*

After this explaining task, students again rated their effort and difficulty during explaining or retrieving and provided a monitoring judgment. Additionally, they rated their enjoyment and interest in the learning activity and students who generated an explanation further stated their perceived social presence during explaining. Finally, all students answered the posttests. The study took approximately 1 hour and was rewarded with 8 €.

## Results

We used partial $\eta_{\text{p}}^2$ and Cohens' $d$ as effect size measures, qualifying values of $\eta_{\text{p}}^2 = .01,.06,.14$ and $d = .20,.50,.80$ as small, medium, and large effects [77]. Additionally, we applied an alpha level of $\alpha = .05$.

### Preliminary analyses

Preliminary analyses showed no significant differences between gender among conditions, $\chi^2$ (6, 137) = 5.50, $p = .538$, $\eta_{\text{p}}^2 = .07$. Results of a MANOVA showed no significant differences between age, students' number of semesters, grade in their last report card in biology, prior knowledge, self-efficacy in explaining, and biological interest across conditions, $F(3, 133) =$ 1.08, $p = .370$. Additionally, students rated the social presence as comparably across the three explaining conditions, $F(2, 100) = 2.23$, $p = .113$, $\eta_{\text{p}}^2 = .04$.

### Learning outcome

The descriptive statistics (see Table 1) suggested that students showed comparable learning outcomes across conditions. Therefore, although our preregistered analysis plan was to conduct two separate contrast analyses for each dependent variable (which would have resulted in four tests in total), we decided to use MANCOVAs instead to reduce the number of statistical tests and to use the most economic statistical approach [78,79]. The text-based and the inference questions were the dependent variables, experimental condition the independent variable, and prior knowledge the control variable. Results showed a main effect of prior

**Table 1. Summary of means and standard deviations (in parentheses) for all measurements.**

|  | Retrieval Condition | Written Condition | Chat Condition | Oral Condition |
|---|---|---|---|---|
| **Learning outcomes** | | | | |
| Text-based questions (0–6) | 3.29 (1.05) | 2.94 (1.12) | 2.77 (1.15) | 3.35 (0.97) |
| Inference questions (0–6) | 2.79 (1.23) | 3.09 (1.09) | 2.89 (1.18) | 2.75 (0.98) |
| **(Meta-)Cognitive processes** | | | | |
| Monitoring accuracy | 0.11 (0.19) | 0.14 (0.22) | 0.17 (0.20) | 0.09 (0.18) |
| Perceived mental effort (1–9) | 5.24 (2.09) | 6.71 (1.36) | 6.66 (1.47) | 6.09 (1.73) |
| Subjective difficulty (1–9) | 4.24 (2.20) | 4.65 (1.84) | 4.94 (2.07) | 5.18 (2.22) |
| **Motivation during learning activity** | | | | |
| Enjoyment (1–4) | 2.29 (0.75) | 2.91 (0.67) | 3.14 (0.73) | 2.91 (0.87) |
| Interest (1–4) | 2.12 (0.71) | 2.74 (0.67) | 3.17 (0.69) | 2.84 (0.79) |
| **Characteristics of explanations** | | | | |
| Personal references | - | 0.12 (0.48) | 1.89 (1.97) | 0.71 (1.66) |
| Concepts | - | 24.47 (5.70) | 21.54 (6.18) | 28.53 (11.39) |
| Elaborations | - | 1.09 (1.03) | 0.94 (0.87) | 1.68 (1.63) |
| **Control variables** | | | | |
| Prior-knowledge (0–5) | 2.06 (1.17) | 2.26 (1.03) | 2.03 (1.18) | 2.25 (1.15) |
| Interest in biology (1–4) | 2.69 (0.90) | 2.54 (0.87) | 2.57 (0.79) | 2.97 (0.74) |
| Self-efficacy explaining (1–4) | 2.83 (0.57) | 2.59 (0.62) | 2.73 (0.65) | 2.80 (0.59) |
| Perceived social presence (1–9) | - | 5.78 (1.80) | 5.73 (2.04) | 4.88 (2.10) |

knowledge, $F(1, 132) = 14.59$, $p < .001$, $\eta_p^2 = .10$, but no differences between conditions, $F(3, 132) = 2.03$, $p = .062$, $\eta_p^2 = .04$ (text-based questions: $F(3, 132) = 2.56$, $p = .058$, $\eta_p^2 = .05$; inference questions: $F(3, 132) = 0.69$, $p = .562$, $\eta_p^2 = .01$). As recommended by Biel and Friedrich [80], to investigate whether the non-significant finding can be attributed to the fact that that the null-hypothesis was true (i.e., no differences among conditions), we additionally computed Bayesian factors with the *bain* package [81]. Values between 0 and 1 can be interpreted as evidence in favor of the alternative hypothesis. Values greater than 1 are suggested as evidence in favor of the null hypothesis. A value of 1 represents no preference for either hypothesis. Results showed a Bayes factor higher than 1 regarding both posttests (text-based questions: $BF_{01} = 9.43$; inference questions: $BF_{01} = 128.04$), suggesting that we can assume that the null-hypothesis is true and that all conditions resulted in comparable learning outcomes.

### Influence of learning activity on (meta-)cognitive processes

**Monitoring accuracy.** In a first step, we conducted one-sample t-tests to investigate whether students significantly over- or underestimated their comprehension after the learning activity (i.e., bias). We contrasted students' actual judgements of their current understanding relative to their performance with an accurate judgment, which was represented by a value of zero. Results revealed that students in all conditions significantly overestimated their current understanding (retrieval condition: $t(33) = 3.43$, $p = .002$, $d = .58$; written condition: $t(33) = 3.74$, $p < .001$, $d = .64$; chat condition: $t(33) = 5.09$, $p < .001$, $d = .85$; oral condition: $t(33) = 2.81$, $p = .008$, $d = .50$).

In a second step, we analyzed potential differences among conditions. We conducted an ANCOVA with experimental condition as independent variable, students' judgements of their current understanding after the learning activity as dependent variable, and students' judgements of their understanding after the study phase as covariate to control for potential intra-

**Table 2. Correlations with confidence intervals regarding all variables.**

| | | 1 | 2 | 3 | 4 | 5 | 6 | 7 | 8 | 9 | 10 | 11 |
|---|---|---|---|---|---|---|---|---|---|---|---|---|
| 1 | Prior knowledge | | | | | | | | | | | |
| 2 | Text-based questions | **.31** [.15,.45] | | | | | | | | | | |
| 3 | Inference questions | **.39** [.24,.52] | **.36** [.20,.50] | | | | | | | | | |
| 4 | Monitoring accuracy | -.14 [-.30,.03] | **-.27** [-.42, -.11] | **-.32** [-.46, -.16] | | | | | | | | |
| 5 | Perceived mental effort | .05 [-.12,.22] | .09 [-.08,.25] | **.19** [.02,.35] | .16 [-.01,.32] | | | | | | | |
| 6 | Subjective difficulty | -.04 [-.21,.13] | -.12 [-.28,.05] | **-.18** [-.34, -.01] | **-.36** [-.50, -.21] | .07 [-.10,.24] | | | | | | |
| 7 | Enjoyment | .05 [-.12,.21] | .09 [-.08,.25] | **.17** [.00,.33] | **.21** [.05,.37] | **.34** [.18,.48] | **-.25** [-.40, -.08] | | | | | |
| 8 | Interest | .02 [-.15,.19] | -.02 [-.19,.15] | .08 [-.09,.24] | **.25** [.08,.40] | **.28** [.11,.42] | **-.18** [-.33, -.01] | **.81** [.75,.86] | | | | |
| 9 | Social presence | .08 [-.11,.27] | -.02 [-.21,.17] | .04 [-.15,.24] | **.21** [.01,.38] | **.28** [.09,.45] | **-.28** [-.45, -.09] | **.47** [.30,.61] | **.40** [.23,.55] | | | |
| 10 | Personal references | .05 [-.12,.21] | .02 [-.15,.19] | .06 [-.11,.23] | .02 [-.15,.19] | **.17** [.01,.33] | .10 [-.07,.26] | **.17** [.00,.33] | .16 [-.01,.32] | .18 [-.01,.36] | | |
| 11 | Concepts | **.34** [.15,.50] | **.45** [.28,.59] | **.34** [.15,.50] | -.03 [-.22,.16] | **.20** [.01,.38] | **-.24** [-.42, -.05] | .10 [-.10,.29] | .05 [-.15,.24] | .07 [-.12,.26] | .07 [-.12,.26] | |
| 12 | Elaborations | **.28** [.09,.45] | **.30** [.11,.47] | **.20** [.00,.38] | .02 [-.17,.21] | .16 [-.04,.34] | **-.26** [-.43, -.07] | .18 [-.01,.36] | .13 [-.07,.31] | **.21** [.02,.39] | .05 [-.15,.24] | **.70** [.59,.79] |

Significant correlations are highlighted in bold font; $p < .05$.

individual differences [4,82]. Results showed a main effect of students' judgements after the study phase, $F(1, 132) = 136.47$, $p < .001$, $\eta_p^2 = .50$, but not of experimental condition, $F(3, 132) = 2.41$, $p = .070$, $\eta_p^2 = .03$. Again, we conducted a Bayesian analysis. Results showed a Bayes factor of $BF_{01} = 51.68$, indicating that all conditions resulted in comparable (biased) judgements.

**Cognitive load.** First, to analyze whether experimental conditions differed regarding students' mental effort during the learning activity, we performed an ANOVA with mental effort as dependent variable and experimental condition as independent variable. Results showed a significant difference among conditions, $F(3, 133) = 5.63$, $p = .001$, $\eta_p^2 = .11$. Post-hoc comparisons (Scheffé) revealed that students who wrote an explanation investigated more mental effort than students who retrieved the material (standard written condition: $p = .006$; chat condition: $p = .008$). The remaining group comparisons were not significant. See Table 2 for correlations with students' learning outcomes.

Second, to investigate potential differences among conditions regarding students' subjective difficulty, we conducted an ANOVA with subjective difficulty as dependent variable and experimental condition as independent variable. Results indicated no differences among conditions, $F(3, 133) = 1.29$, $p = .282$, $\eta_p^2 = .03$. Bayesian analyses showed a Bayes factor of $BF_{01} = 44.79$, indicating that there were no differences between conditions regarding students' subjective difficulty.

## Motivation during the learning activity

**Enjoyment.** We explored differences regarding students' enjoyment during the learning activity by conducting an ANOVA with experimental condition as independent variable.

Results revealed significant differences among conditions, $F(3, 133) = 7.91$, $p < .001$, $\eta_p^2 = .15$. Post-hoc comparisons showed that students in all explaining conditions (i.e., standard written condition: $p = .012$; chat condition: $p < .001$; oral condition: $p = .012$) enjoyed the learning activity more than students in the retrieval condition.

**Interest.** Similarly, we investigated potential differences regarding the interestingness of the learning activity among conditions. Results of an ANOVA with experimental condition as independent variable showed a significant effect of condition, $F(3, 133) = 13.11$, $p < .001$, $\eta_p^2 = .23$. Post-hoc comparisons revealed that students rated all three explaining conditions (i.e., standard written condition: $p = .006$; chat condition: $p < .001$; oral condition: $p < .001$) as more interesting than the retrieval condition.

### Characteristics of the explanations

As expected, within the explaining conditions, the generated explanations differed among conditions: Personal references, $F(2, 100) = 12.19$, $p < .001$, $\eta_p^2 = .20$; concepts, $F(2, 100) = 6.38$, $p = .002$, $\eta_p^2 = .11$, and elaborations, $F(2, 100) = 10.35$, $p = .034$, $\eta_p^2 = .07$. Post-hoc comparisons (Scheffé) showed that students mentioned more personal references in the chat condition compared to the standard written condition, $p < .001$, and the oral condition, $p < .001$, which can be regarded as an indicator that the chat condition indeed induced higher levels of social presence. Additionally, students who explained orally mentioned more concepts, $p = .003$, and provided more elaborations, $p = .048$, than students in the chat condition. None of the other comparisons were significant.

### Discussion

The aim of the current study was to investigate whether increasing social presence during writing explanations aids learning. Contrarily to our preregistered hypotheses, we did not obtain a significant effect of induced social presence during the learning activity compared to a less social learning environment. Apparently, only raising the social presence did not contribute to learning. Additionally, we did not find significant differences between the explaining and the retrieval conditions. Thus, we did not replicate prior findings on the effectiveness of explaining [1,9,22,23,83]. Relatedly, we were also not able to replicate beneficial effects of oral explaining compared to writing explanations [11–13]. The null findings regarding students' comprehension might be attributed to low levels of prior knowledge. Little prior knowledge limits students in learning new contents adequately [30,84]. However, prior research revealed that generative learning activities, such as explaining, are in particular beneficial for low prior knowledge students [9]. Additionally, students showed comparable monitoring accuracy ratings across conditions. Interestingly, all students overestimated their current understanding. This, however, is in line with prior research that highlighted that students generally tend to overestimate their current comprehension [85–87]. Nevertheless, we need to reject our hypotheses regarding effects on students' comprehension and their monitoring accuracy. We want to note, however, that the obtained null-findings are in line with a growing body of empirical research that was also not able to document an effect of explaining [3,4,8,88].

In this context, our study is of particular interest, as we used previously tested materials and knowledge tests in the context of explaining [13], and had sufficient test power to test our hypotheses. As additional safe-guard we also computed Bayes factors, which similarly suggested that our findings are rather in favor of the null-hypothesis. Thus, we are confident that this study represents reliable evidence to reject the explaining and modality hypothesis, at least in the current study context.

Furthermore, the analyses of the characteristics of the explanations demonstrated that the quality of the generated explanations was rather low, suggesting that the students were less capable to generate effective explanations. This finding may explain why we did not find any effects on learning. Our explorative analyses regarding the (meta-)cognitive and motivational conditions of explaining revealed distinct differences between explaining and retrieval, as students perceived more effort and more motivation (i.e., enjoyment, interest) during the learning activity, which is in line with previous research [13,23]. Cognitive conditions, such as mental effort, are strongly linked to students' learning [89], and, therefore, should be investigated in more detail in combination with additional learning activities, such as learning by explaining, in future research. Moreover, learning enjoyment and relatedly interest are regarded as important value-related facets of intrinsic motivation [55], and intrinsic motivation is particularly key to whether and how students would persist to use explaining as a learning strategy beyond the experimental context [90]. To explore the differences regarding students' enjoyment and mental effort in more detail, we additionally conducted explorative mediation analyses by applying a bootstrapping approach with 1,000 simulations based on Hayes [91]. Results revealed that enjoyment but not mental effort mediated learning by explaining on students' text-based comprehension as the indirect effect was significant (ACME = 0.24, 95% CI [0.03, 0.50], $p$ = .024). These findings are in line with the results by Hoogerheide and colleges who demonstrated that explaining led to higher levels of enjoyments, which, in turn, enhanced students' learning [23].

Nevertheless, against the background of the relatively low quality of students' explanations, as a theoretical consequence of our study, we suggest that future research should focus on how students could be supported to generate high-quality explanations. Given that students are generally not familiar with explaining challenging contents [92], they might depend on further support to be able to apply this learning activity successfully. As a first attempt, Lachner and Neuburg [93] investigated the role of formative feedback during explaining (in written form). They found that formative feedback helped students generate more cohesive explanations, which finally contributed to their comprehension. Additionally, the self-explaining literature focusses on two main instructional approaches [94], supporting self-explaining directly by means of pre-trainings [41] or indirectly by the use of prompts [39,95], which should work as strategy activators to enact deep-level explaining strategies. Whether and under which conditions such direct and indirect support procedures also hold true for generating explanations to fictitious others has to be investigated by further research.

## Study limitations and future research

With our study, we provide the first empirical approach to systematically investigate the influence of perceived social presence on the effectiveness of explaining to fictitious others. In this context, however, we would like to point out some limitations of our study. First, an unexpected finding was that we indeed found an effect of our intervention on the number of personal references, but not on the social presence ratings. This finding might be contradictory at first glance. However, we want to note that such contradictory findings frequently occur in the context of inducing discourse situations [96,97]. As such, these findings are often interpreted as suggesting that inducing discourse situation may not affect subsequent communication processes directly, but rather subconsciously. This may explain the differences between the number of personal references and the intentional presence ratings.

Another surprising finding was that students in the explaining conditions did not outperform students in the retrieval practice condition. This not only contradicts our assumptions but also prior studies that showed beneficial effects of explaining in contrast to restudying or

retrieving the learned material [1,2,4,12,13,23]. In this context, we would like to point out that we used the same learning materials as in a previous study by Jacob, Lachner, and Scheiter who obtained effects of explaining with this difficult learning material [13]. In contrast to Jacob and colleges, students in our study showed slightly lower learning outcomes. Additionally, results indicated a higher correlation between prior knowledge and students' learning outcome, suggesting that the provided learning material was too difficult, which may explain the null findings. Seemingly, students have problems in applying this learning strategy successfully when the learning material is overly difficult [30], which highlights the need for additional support during learning, such as pre-trainings or prompts [39,43,95].

Another caveat refers to the measurement of the generated explanations. Based on prior research, we measured the number of personal references, concepts, and elaborations in students' generated explanations [11–13,58,70,71,98]. We used these characteristics as a coarse proxy for the underlying processes during the learning activity [2,11,13]. Further research is needed to explore the generated explanations in more detail which could provide a deeper insight into the explaining effect. This indeed could disentangle which detrimental effects occurred during explaining. Further research should, therefore, implement additional online-measures, for instance, think-aloud protocols or log-file-data [99].

Another limitation refers to the relatively low homogeneity indices. However, we want to note that we measured a broad understanding of the topic by asking different and independent questions with a restricted set of 5 to 6 items per test. This decision likely resulted in a trade-off regarding the internal consistency of the knowledge measures, as they likely covered diverse sub-components of students' comprehension [100]. Further research with advanced statistical procedures is needed which explicitly takes the multidimensionality into account, for instance, by using bifactor-(S-1) measurement models which, however, require larger sample sizes [101].

Finally, we would like to point out restrictions regarding the measurement of enjoyment, as we only measured enjoyment quantitatively by means of two items. Our results revealed, in line with the study by Hoogerheide and colleges [23], that explaining resulted in higher levels of enjoyment compared to retrieving the materials, which is in part indicative for the prognostic validity of our instrument. However, as we only used a quantitative measure, we are not able to conclude whether and how enjoyment affected learning outcomes in a qualitative manner. Therefore, further research is needed that investigates the impact of enjoyment on the explaining effect in more detail by implementing qualitative methods, such as interviews [102,103].

## Conclusion

The findings of our study show that explaining does not necessarily foster learning, although it enhances investments of mental effort, and increases students' motivation. Furthermore, our findings indicate that simply inducing social presence does not increase the effectiveness of explaining. It is, therefore, up for further research to find ways to trigger deep-learning processes during explaining. As such, explaining can contribute to students' mental effort and motivation, which are also linked to higher learning outcomes.

## Supporting information

**S1 Dataset.**
(SAV)

## Acknowledgments

We would like to thank Eleonora Dolderer, Lisa Holzschuh, Gamze Coecen, and Louisa Döderlein for transcribing and rating the qualitative data. Leonie Jacob is a doctoral student at the LEAD Graduate School & Research Network [GSC1028], which was funded within the framework of the Excellence Initiative of the German federal and state governments.

## Author Contributions

**Conceptualization:** Leonie Jacob, Andreas Lachner, Katharina Scheiter.

**Data curation:** Leonie Jacob.

**Formal analysis:** Leonie Jacob, Andreas Lachner.

**Funding acquisition:** Katharina Scheiter.

**Investigation:** Leonie Jacob.

**Methodology:** Leonie Jacob, Andreas Lachner.

**Project administration:** Leonie Jacob.

**Supervision:** Andreas Lachner, Katharina Scheiter.

**Writing – original draft:** Leonie Jacob.

**Writing – review & editing:** Andreas Lachner, Katharina Scheiter.

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
