## [Decision Letter · Decision Letter 0]

12 Feb 2021

PONE-D-20-39683

Does increasing social presence enhance the effectiveness of writing explanations?

PLOS ONE

Dear Dr. Jacob,

Thank you for submitting your manuscript to PLOS ONE. After careful consideration, we feel that it has merit but does not fully meet PLOS ONE’s publication criteria as it currently stands. Therefore, we invite you to submit a revised version of the manuscript that addresses the points raised during the review process

Please pay special attention to the two major points raised by Reviewer 1: (1) There is need for describing social presence; and (2) As the three explaining conditions reported similar levels of social presence, how could there be any effects attributed to social presence? This last point can be included in the Discussion.   

We look forward to receiving your revised manuscript.

Kind regards,

Juan Cristobal Castro-Alonso, Ph.D.

Academic Editor

PLOS ONE

Journal Requirements:

2) If materials, methods, and protocols are well established, authors may cite articles where those protocols are described in detail, but please ensure that your submission includes sufficient information to be understood independent of these references (https://journals.plos.org/plosone/s/submission-guidelines#loc-materials-and-methods).

3)  We note that the grant information you provided in the ‘Funding Information’ and ‘Financial Disclosure’ sections do not match.

4) Please include captions for your Supporting Information files at the end of your manuscript, and update any in-text citations to match accordingly. Please see our Supporting Information guidelines for more information: http://journals.plos.org/plosone/s/supporting-information.

5) We note that Figure 1 includes an image of a participant in the study. 

Reviewers' comments:

Reviewer's Responses to Questions

**Comments to the Author**

1. Is the manuscript technically sound, and do the data support the conclusions?

Reviewer #1: Yes

Reviewer #2: Yes

2. Has the statistical analysis been performed appropriately and rigorously? 

Reviewer #1: Yes

Reviewer #2: Yes

3. Have the authors made all data underlying the findings in their manuscript fully available?

Reviewer #1: Yes

Reviewer #2: Yes

4. Is the manuscript presented in an intelligible fashion and written in standard English?

Reviewer #1: Yes

Reviewer #2: Yes

5. Review Comments to the Author

Reviewer #1: Thank you for the opportunity to review this manuscript, which I think is methodologically sound, well written, and overall good work. In terms of methodology, I especially liked that your hypotheses were pre-registered, you conducted a power analysis, and that you complemented your analyses with Bayes Factor. Substantively, I think your work is an important contribution and I like that you explored social presence as an avenue for effective learning via explanation.

I only have two major points of substantive critique that should be discussed to improve the manuscript, as well as some minor points.

Major point #1:

The manuscript is in need of a review of the concept of social presence. It need not be very long but you should, in my opinion, spend a few lines on what is meant by social presence (psychologically) and what is hoped to be the benefit (educationally). Also, although I'm not asking you to wildly speculate about the mechanism that is expected to bring about learning benefits, the manuscript may be improved by some explanation of how and why this social concept could interact with cognitive factors to account for improvements in learning. Generally, looking at the literature section, I find there to be too few references to social presence considering that the construct is central to your study.

Major point #2:

The second critique I have is a logical one. One could argue that expecting students to experience a sense of social presence with a fictitious peer goes against the very notion of social presence itself, which refers to experiencing a sense of non-mediation with another *real* human (despite the setting being, in fact, mediated). This could be discussed in the limitations section. In my opinion, this contradiction is also evidenced in the results that there was no difference in social presence between conditions. The seemingly conflicting results of similar SP across conditions but different amounts of personal references across conditions is, in my opinion, consistent with no actual difference in SP and just varying communication habitus in different social situations. So one could argue that the manipulation check has failed and I think this could be discussed a bit more critically.

Minor points:

- I think you overstate the legitimacy of the behavioral measure of SP (p.25, top). More commonly used are self-report measures and although this too has limitations, we can assume that they measure social presence whereas the personal-reference method could also measure some other interpersonal aspect of social interaction. Looking at the references you provide here, I do not see evidence for the validity of personal references for measuring SP. Please correct me if I'm wrong.

- On page 17, you write that you deviated from the analysis plan by conducting a MANCOVA instead of two separate contrast analyses "to reduce the number of statistical tests". I don't understand this rationale. Although this might be due to a gap of knowledge on my part, maybe you could just say a bit more about that, as deviations from pre-registered plans should be well-argued.

- On page 6 and 7, there is talk about differential effects of explanation vs retrieval, depending on difficulty of learning material. In interpreting these findings, I think the authors make the mistake of concluding significant differences between a statis. significant result and a statist. non-significant result. See Gelman & Stern 2006 "The Difference Between “Significant” and “Not Significant” is not Itself Statistically Significant". Please have a look at this. If I have misinterpreted, please ignore my objection.

That said, I think your manuscript reports on valuable and sound work. I hope you find my comments helpful in further improving the manuscript.

Reviewer #2: In general, the authors conducted a technically sound study. However, I have some concerns regarding the theoretical underpinnings of their study. The theoretical part of their manuscript is rather descriptive and presents primarily results of empirical studies. In addition a lot of constructs (e.g., learning, enjoyment, complexity of the learning material, monitoring accuracy, mental effort, subjective difficulty) are mentioned whose relationships seems a little bit unclear. Therefore, it would be better to provide a theoretical model from which the hypotheses are clearly derived. For example, that social presence should enhance the effectiveness of explanations is explained with the argument that writing alone should lead to a knowledge-telling bias. However, one might easily provide the counter-argument that writing is good because learners have sufficient time to adopt a knowledge-transforming stance. In addition, the authors should present their hypotheses more thoroughly in the context of the specifics of their study (e.g., which persons, which learning material).

In addition, I have the following comments:

Was the learning material too difficult because the means in learning outcomes revolve around the theoretical mean of 3 correctly answered questions? Can this explain the null effects?

Why did the authors use a different number of points for the different Likert scales (e.g., interest from 1 to 4, social presence form 1 to 9)?

Please mention whether the intercept in the analysis of monitoring accuracy was significant across conditions. This would reveal whether the overestimation was significant or not. Why were learners rather accurate in their judgments?

Even though enjoyment is significantly lower in the retrieval condition, is it legitimate to conclude that there is really a qualitatively different form of enjoyment?

6. PLOS authors have the option to publish the peer review history of their article (what does this mean?). If published, this will include your full peer review and any attached files.

Reviewer #1: **Yes: **Joshua Weidlich

Reviewer #2: No

---

## [Author Response · Author response to Decision Letter 0]

25 Feb 2021

Reviewer #1

Thank you for the opportunity to review this manuscript, which I think is methodologically sound, well written, and overall good work. In terms of methodology, I especially liked that your hypotheses were pre-registered, you conducted a power analysis, and that you complemented your analyses with Bayes Factor. Substantively, I think your work is an important contribution and I like that you explored social presence as an avenue for effective learning via explanation.

Thank you very much for your appreciative words. 

I only have two major points of substantive critique that should be discussed to improve the manuscript, as well as some minor points.

We considered both points accordingly.

Major point #1:

The manuscript is in need of a review of the concept of social presence. It need not be very long but you should, in my opinion, spend a few lines on what is meant by social presence (psychologically) and what is hoped to be the benefit (educationally). Also, although I'm not asking you to wildly speculate about the mechanism that is expected to bring about learning benefits, the manuscript may be improved by some explanation of how and why this social concept could interact with cognitive factors to account for improvements in learning. Generally, looking at the literature section, I find there to be too few references to social presence considering that the construct is central to your study.

We agree that we introduced the concept of social presence rather briefly. In the revised version, we therefore now provide a thorough definition of social presence based on Gunawardena & Zittle, 1997, Oh et al., 2018, and Short et al., 1976 (see page 3):

“Social presence is a central concept in discourse theory, and is commonly defined as the extent to which a person feels that a communication partner is present during a mediated conversation, such as in online learning environments [14-18]. In this context, social presence not only holds true for real persons, but also for virtual or even fictitious communication partners [19-21]."

Additionally, as also requested by Reviewer 2 (see first comment), we now provide a theoretical model for two potential mechanisms of social presence during learning by explaining on page 6 (see Lachner, Jacob, & Hoogerheide, 2021, and Hoogerheide, Visee, Lachner, van Gog, 2019):

“Even though serval studies indicated beneficial effects of learning by explaining to fictitious others on students’ comprehension and monitoring skills, little is yet known about the underlying mechanism of why explaining is effective. Recently, researchers emphasized the role of social presence during explaining [11-13, 47], as higher levels of social presence (which also may arise from virtual of even fictitious characters) are linked to central components of learning, such as cognition or motivation [14, 18-21, 48, 49]. On the one hand, the social presence of a fictitious communication partner may engage students to adapt their knowledge to the audience’s needs [50, 51], for instance, by providing further details and elaborations that go beyond the contents of the learning material. Such audience-adjustments may result in deeper elaborative processes and contribute to meaningful learning [52]. On the other hand, from a motivational perspective [23], the social presence of a fictitious person may also increase the feeling of relatedness [53]. Higher levels of relatedness may yield higher levels of enjoyment and investments during providing an explanation, and, in turn, contribute to comprehension [23, 53, 54].”

Major point #2:

The second critique I have is a logical one. One could argue that expecting students to experience a sense of social presence with a fictitious peer goes against the very notion of social presence itself, which refers to experiencing a sense of non-mediation with another *real* human (despite the setting being, in fact, mediated). This could be discussed in the limitations section. In my opinion, this contradiction is also evidenced in the results that there was no difference in social presence between conditions. The seemingly conflicting results of similar SP across conditions but different amounts of personal references across conditions is, in my opinion, consistent with no actual difference in SP and just varying communication habitus in different social situations. So one could argue that the manipulation check has failed and I think this could be discussed a bit more critically.

Given that social presence is commonly operationalized as a continuous and not a binary construct, we would argue that social presence can also occur in fictitious communication settings. Based on this assumption, the chat condition should elicit higher levels of social presence than the two explaining conditions, and a self-referential learning activity, such as retrieval practice. Relatedly, previous research documented that social presence also occurs in virtual settings (e.g., Atkinson, 2002; Kim, 2013; Wang & Antonenko, 2017). The chat condition should show the highest levels of social presence for three reasons. First, we provided different social cues, such as a picture and the name of the communication partner. Second, to increase the feeling of being together, we also implemented the explaining task in a simulated WhatsApp Chat to evoke a computer-mediated, but direct communication situation. Third, we also aligned the time of the chat to emphasize the perceived synchrony of the communication. We would therefore still argue that we varied the social presence in our study. However, we admit that the manipulation should be described in more detail. We therefore provide a more comprehensive presentation of our manipulation in the manuscript (see page 17).

“We provided different social cues to induce higher levels of social presence (see Weidlich & Bastiaens for a similar approach; 18). First, students could see a profile picture from Lisa. Second, they received a short message from Lisa who directly asked the students for an explanation (Fig 1). We automatically adapted the time of the received message to increase the synchrony of the communication. Students could send Lisa a message to share their explanation within the chat.”

Regarding the last comment, we are still convinced that personal references may constitute valid measures of social presence (see also Hoogerheide et al., 2016; Jacob et al., 2020; Lachner et al., 2021 for recent applications), since they have been frequently used in discourse analysis as proxies for social presence (e.g., Chafe, 1982; Chafe & Tannen, 1987; Akinnaso, 1985; Einhorn, 1978). That said, the discrepancy between the findings of social presence ratings and personal references does not necessarily mean that our manipulation failed (as indicated by the differences in personal references), but rather that the feeling of social presence does not necessarily have to be an intentional, but rather a subconscious process evoked by our manipulation (see Weinhuber et al., 2019; Savary et al., 2015, for related evidence in discourse research). Nevertheless, we agree that this issue needs to be discussed and included it in our limitations in the discussion section (see page 27): 

“With our study, we provide the first empirical approach to systematically investigate the influence of perceived social presence on the effectiveness of explaining to fictitious others. In this context, however, we would like to point out some limitations of our study. First, an unexpected finding was that we indeed found an effect of our intervention on the number of personal references, but not on the social presence ratings. This finding might be contradictory at first glance. However, we want to note that such contradictory findings frequently occur in the context of inducing discourse situations [89, 90]. As such, these findings are often interpreted as suggesting that inducing discourse situation may not affect subsequent communication processes directly, but rather subconsciously. This may explain the differences between the number of personal references and the intentional presence ratings.”

Minor points:

- I think you overstate the legitimacy of the behavioral measure of SP (p.25, top). More commonly used are self-report measures and although this too has limitations, we can assume that they measure social presence whereas the personal-reference method could also measure some other interpersonal aspect of social interaction. Looking at the references you provide here, I do not see evidence for the validity of personal references for measuring SP. Please correct me if I'm wrong.

We already argued in the previous comment why we think that the number of personal references may be an appropriate means to measure social presence. The intention to use personal references was based on prominent discourse literature, which regularly uses personal references as indicators of social presence (Akinnaso, 1985; Chafe, 1982; Chafe & Tannen, 1987; Sindoni, 2013; Tannen, 1982). That said the concept of personal references is frequently used in the learning by explaining literature (see Jacob et al., 2020; Lachner et al., 2018). Nevertheless, we agree that the argumentation for why we used social presence was rather brief. We revised the manuscript accordingly (see page 14):

“Based on discourse literature, we used personal references (i.e., “I”, “you”, etc.) as an indicator for the perceived social presence [55, 56, 68, 69] which is frequently used in learning by explaining literature [11-13]. We automatically counted the number of personal references with RStudio [70].”

As stated before, we additionally discussed the measure of social presence in the limitation section on page 27:

“With our study, we provide the first empirical approach to systematically investigate the influence of perceived social presence on the effectiveness of explaining to fictitious others. In this context, however, we would like to point some limitations of our study. First, an unexpected finding was that we indeed found an effect of our intervention on the number of personal references, but not on the social presence ratings. This finding might be contradictory at first glance. However, we want to note that such contradictory findings frequently occur in the context of inducing discourse situations [89, 90]. As such, these findings are often interpreted that way, that inducing discourse situation may not affect subsequent communication processes directly, but rather subconsciously, which may explain the differences between the number of personal references and the intentional presence ratings.”

- On page 17, you write that you deviated from the analysis plan by conducting a MANCOVA instead of two separate contrast analyses "to reduce the number of statistical tests". I don't understand this rationale. Although this might be due to a gap of knowledge on my part, maybe you could just say a bit more about that, as deviations from pre-registered plans should be well-argued.

Indeed, we preregistered to run two separate contrast analyses for each dependent variable, resulting in a total of four analyses. However, after inspecting the descriptive data it was obvious that the four conditions showed very similar scores regarding students’ comprehension. In line with the statistical aim to use the most economic and parsimonious method as possible (Ranganathan, Pramesh, & Buyse, 2016; Streiner, 2015), we decided to use MANCOVAs instead. We added the reason for our decision in the revised version accordingly (see page 19):

“The descriptive statistics (see Table 1) suggested that students showed comparable learning outcomes across conditions. Therefore, although our preregistered analysis plan was to conduct two separate contrast analyses for each dependent variable (which would have resulted in four tests in total), we decided to use MANCOVAs instead to reduce the number of statistical tests and to use the most economic statistical approach [76, 77].”

- On page 6 and 7, there is talk about differential effects of explanation vs retrieval, depending on difficulty of learning material. In interpreting these findings, I think the authors make the mistake of concluding significant differences between a statis. significant result and a statist. non-significant result. See Gelman & Stern 2006 "The Difference Between “Significant” and “Not Significant” is not Itself Statistically Significant". Please have a look at this. If I have misinterpreted, please ignore my objection.

We rewrote the sentence accordingly (see pages 7-8):

“Results revealed an interaction effect between learning activity and text difficulty (d = 0.31). Additional contrast analyses showed that the explaining effect only held true when students learned from difficult but not from less difficult material, as they found a significant effect of explaining in the high-difficult condition, but not in the low-difficult condition.”

Reviewer #2

In general, the authors conducted a technically sound study.

Thank you very much.

However, I have some concerns regarding the theoretical underpinnings of their study. The theoretical part of their manuscript is rather descriptive and presents primarily results of empirical studies. In addition a lot of constructs (e.g., learning, enjoyment, complexity of the learning material, monitoring accuracy, mental effort, subjective difficulty) are mentioned whose relationships seems a little bit unclear. Therefore, it would be better to provide a theoretical model from which the hypotheses are clearly derived. For example, that social presence should enhance the effectiveness of explanations is explained with the argument that writing alone should lead to a knowledge-telling bias. However, one might easily provide the counter-argument that writing is good because learners have sufficient time to adopt a knowledge-transforming stance. In addition, the authors should present their hypotheses more thoroughly in the context of the specifics of their study (e.g., which persons, which learning material).

We added a theoretical model accordingly. First, we grounded the literature of learning-by-explaining in the generative learning literature (e.g., Fiorella & Mayer, 2015), and included an extended version of the theoretical framework (see pages 4-5):

“Learning by explaining is a generative learning activity which aims at enhancing students’ meaningful learning [27]. In line with the generative learning theory, the process of explaining as a generative act may elicit cognitive (e.g., mental effort) and metacognitive (e.g., monitoring) processes, which should contribute to students’ comprehension: First, and in line with Mayer’s SOI model [28], when explaining students need to select the most relevant information of the provided materials and to organize the information in a coherent way. Then, they need to connect the new contents with their already existing knowledge to integrate them into their long-term memory [27, 29, 30]. Through this connection, students are able to provide explanations that include further details and information that go beyond the giving materials, which results in new knowledge and meaningful learning [27-34]. This process additionally triggers students to monitor whether they understood all relevant contents correctly or whether they need to restudy specific information. As a consequence, students’ metacognitive monitoring may become more accurate when explaining, which has also been observed for other generative learning activities such as keyword generation or gap filling [3, 4, 35, 36]. Learning by explaining is commonly implemented in interactive learning settings in which students explain learned contents to present and interactive peers; this setting allows students to exchange ideas and thought, which additionally enhances their understanding [6, 7, 31, 35-45]. Interestingly, recent research started to investigate the effectiveness of explaining to a fictitious peer and reported promising results [1, 2, 4, 12, 13, 23].”

More importantly, we provide a theoretical model for why learning by explaining to fictitious others is effective, and introduce social presence as central mechanism (see also Reviewer 1, major point #1). In this context, we discussed two potential mechanisms of social presence during learning by explaining (see Lachner, Jacob, & Hoogerheide, 2021, and Hoogerheide, Visee, Lachner, van Gog, 2019), see page 6 in the revised manuscript: 

“Even though, serval studies indicated beneficial effects of learning by explaining to fictitious others on students’ comprehension and monitoring skills, still little is known about the underlying mechanism of why explaining is effective. Recently, researchers emphasized the role of social presence during explaining [11-13, 47], as higher levels of social presence (which also may arise from virtual of even fictitious characters) are linked to central components of learning, such as cognition or motivation [14, 18-21, 48, 49]. On the one hand, the social presence of a fictitious communication partner may engage students to adapt their knowledge to the audience’s needs [50, 51], for instance by providing further details and elaborations that go beyond the contents of the learning material. Such audience-adjustments may result in deeper elaborative processes and contribute to meaningful learning [52]. On the other hand, from a motivational perspective [23], the social presence of a fictitious person may also increase the feeling of relatedness [53]. Higher levels of relatedness may yield in higher levels of enjoyment and investments during providing an explanation, and contribute to comprehension [23, 53, 54].”

In addition, I have the following comments:

Was the learning material too difficult because the means in learning outcomes revolve around the theoretical mean of 3 correctly answered questions? Can this explain the null effects?

We used the learning material from a previous study by Jacob, Lachner, and Scheiter (2020), who obtained effects of explaining as compared to retrieval practice with the learning material. Indeed, the learning material was rather difficult and challenging. This, however, was indented as the study by Jacob and colleagues demonstrated that effects of explaining were stronger when the material was challenging (see also Lachner et al., 2018). Admittedly, the mean scores were somewhat lower than in the study by Jacob et al. (2020). That said, we also found a slightly higher correlation between prior knowledge and learning outcomes, suggesting that the material could have been even more difficult for our participants of the current study. We now include this point in our discussion section (see pages 27-28): 

“Another surprising finding was that students in the explaining conditions did not outperform students in the retrieval practice condition. This not only contradicts our assumptions but also prior studies that showed beneficial effects of explaining in contrast to restudying or retrieving the learned material [1, 2, 4, 12, 13, 23]. In this context, we would like to point out that we used the same learning materials as in a previous study by Jacob, Lachner, and Scheiter who obtained effects of explaining with this difficult learning material [13]. In contrast to Jacob et al., students in our study showed slightly lower learning outcomes. Additionally, results indicated a higher correlation between prior knowledge and students’ learning outcome, suggesting that the provided learning material was too difficult, which may explain the null findings. Seemingly, students have problems in applying this learning strategy successfully when the learning material is overly difficult [30], which highlights the need for additional support during learning, such as pre-trainings or prompts [37, 41, 88].”

Why did the authors use a different number of points for the different Likert scales (e.g., interest from 1 to 4, social presence form 1 to 9)?

Our scales, as well as the answer formats were mostly based on prior research (Baars et al., 2017; Jerusalem & Schwarzer, 1999; Kunter et al., 2013). Regarding students’ motivation, we followed common approaches on measuring motivation that used 4-point Likert scales (e.g., Hertel, 2009; Lazarides, Gaspard, & Dicke, 2019; Seidel et al., 2011). As an exception, we self-generated the scale for social presence. Such scales are commonly applied in measuring subjective assessments of task characteristics, such as perceived mental effort or subjective difficulty, as they are regarded to be more sensitive towards capturing differences in task characteristics (e.g., DeLeeuw & Mayer, 2008; Paas, 1992). We revised the corresponding paragraph accordingly (see page 15):

“We asked students to rate their feelings of a social presence of the fictitious peer during explaining. We asked students to rate three self-generated items (i.e., “How strongly did you imagine that Lisa was real?”; “How important was it for you that Lisa understands the contents?”; “How strongly did you perceived being in a communicative situation”; McDonald’s ωt = .91). Based on related approaches on subjective assessments of task characteristics, we used a 9-point Likert scale from 1 “not at all” to 9 “completely” [24, 25].”

Please mention whether the intercept in the analysis of monitoring accuracy was significant across conditions. This would reveal whether the overestimation was significant or not. Why were learners rather accurate in their judgments?

if we understood the comment correctly, the reviewer is interested whether students significantly over- or underestimated their current comprehension in the separate conditions. Commonly, research rather uses individual t-tests against zero (which, in our case, represents an accurate judgement) to test whether students significantly differ from accurate judgements within conditions (El Rafaey et al., 2015; Kiebel & Holmes, 2006; Widmer et al., 2016). We therefore followed this literature and conducted one-sample t-tests against zero. Results showed that all condition significantly overestimated their current comprehension. We restructured the corresponding analyses in the results section to include these analyses (see pages 20-21):

“In a first step, we conducted one-sample t-tests to investigate whether students significantly over- or underestimated their comprehension after the learning activity (i.e., bias). We contrasted students’ actual judgements of their current understanding relative to their performance with an accurate judgment, which was represented by a value of zero. Results revealed that students in all conditions significantly overestimated their current understanding (retrieval condition: t(33) = 3.43, p = .002, d = .58; written condition: t(33) = 3.74, p < .001, d = .64; chat condition: t(33) = 5.09, p < .001, d = .85; oral condition: t(33) = 2.81, p = .008, d = .50). In a second step, we analyzed potential differences among conditions. We conducted an ANCOVA with experimental condition as independent variable, students’ judgements of their current understanding after the learning activity as dependent variable, and students’ judgement of their understanding after the study phase as covariate to control for potential intra-individual differences [4, 80]. Results showed a main effect of students’ judgements after the study phase, F(1, 132) = 136.47, p < .001, η²p = .50, but not of experimental condition, F(3, 132) = 2.41, p = .070, η²p = .03. Again, we conducted a Bayesian analysis. Results showed a Bayes factor of BF01 = 51.68, indicating that all conditions resulted in comparable (biased) judgements.”

Even though enjoyment is significantly lower in the retrieval condition, is it legitimate to conclude that there is really a qualitatively different form of enjoyment?

Thank you for your comment. We indeed only measured enjoyment quantitatively, similar to Hoogerheide, Visee, et al. (2019). In their study, the authors reported a mediating effect of enjoyment: Students who explained enjoyed the learning activity more than the control group, which resulted in higher learning outcomes. To investigate the role of enjoyment a step further, we conducted an explorative mediation analysis with experimental condition as the independent variable, students’ learning outcome as the dependent variable and students’ self-reported enjoyment as the mediator variable. Similar to Hoogerheide, Visee, et al. (2019), results revealed a mediating effect of enjoyment, which is indicative for the quality of our assessment. We included this result in our discussion section (see page 26):

“To explore the differences regarding students’ enjoyment and mental effort in more detail, we additionally conducted explorative mediation analyses by applying a bootstrapping approach with 1,000 simulations based on Hayes [85]. Results revealed that enjoyment but not mental effort mediated learning by explaining on students’ text-based comprehension as the indirect effect was significant (ACME = 0.24, 95 % CI [0.03, 0.50], p = .024). These findings are in line with the results by Hoogerheide and colleges who demonstrated that explaining led to higher levels of enjoyments, which, in turn, enhanced students’ learning [23].”

Nevertheless, we admit that we did not measure enjoyment qualitatively, and therefore are not able to provide any qualitative conclusions on our findings. We added this shortcoming as potential limitation of our study (see page 28):

“Finally, we would like to point out restrictions regarding the measurement of enjoyment, as we only measured enjoyment quantitatively by means of two items. Our results revealed, in line with the study by Hoogerheide et al. [23], that explaining resulted in higher levels of enjoyment compared to retrieving the materials, which is in part indicative for the prognostic validity of our instrument. However, as we only used a quantitative measure we are not able to conclude whether and how enjoyment affected learning outcomes in a qualitative manner. Therefore, further research is needed that investigates the impact of enjoyment on the explaining effect in more detail by implementing qualitative methods, such as interviews [93, 94].”

 

References

Atkinson RK. Optimizing learning from examples using animated pedagogical agents. Journal of Educational Psychology 2002; 94(2):416–27.

Akinnaso FN. On the similarities between spoken and written language. Language and Speech 1985; 28:323–59.

Baars M, van Gog T, Bruin A de, Paas F. Effects of problem solving after worked example study on secondary school children’s monitoring accuracy. Educational Psychology 2017; 37(7):810–34.

Chafe W. Integration and involvement in speaking, writing, and oral literature. In: Tannen D, editor. Spoken and written language: Exploring orality and literacy. Norwood, NJ: Ablex; 1982. p. 35–54.

Chafe W, Tannen D. The relation between written and spoken language. Annual Review of Anthropology 1987; 16(1):383–407.

DeLeeuw KE, Mayer RE. A comparison of three measures of cognitive load: Evidence for separable measures of intrinsic, extraneous, and germane load. Journal of Educational Psychology 2008; 100(1):223–34.

El Refaey, M, Watkins, C, P, Kennedy, E. Oxidation of the aromatic amino acids tryptophan and tyrosine disrupts their anabolic effects on bone marrow mesenchymal stem cells. Molecular and Cellular Endocrinology 2015; 410: 87-96.

Einhorn L. Oral and written style: An examination of differences. Southern Speech Communication Journal 1978; 43(3):302–11.

Fiorella L, Mayer RE. Eight ways to promote generative learning. Educ Psychol Rev 2015; 28(4):717–41.

Gunawardena CN, Zittle FJ. Social presence as a predictor of satisfaction within a computer‐mediated conferencing environment. American Journal of Distance Education 1997; 11(3):8–26.

Hertel S, Klieme E. PISA 2009 Skalenhandbuch. Münster: Waxmann; 2014.Lakoff RT. Some of my favorite writers are literate: The mingling of oral and literate strategies in written communication. In: Tannen D, editor. Spoken and written language: Exploring orality and literacy. Norwood, NJ: Ablex; 1982. p. 239–60.

Hoogerheide V, Visee J, Lachner A, van Gog T. Generating an instructional video as homework activity is both effective and enjoyable. Learning and Instruction 2019; 64:101226.

Hoogerheide V, Deijkers L, Loyens SM, Heijltjes A, van Gog T. Gaining from explaining: Learning improves from explaining to fictitious others on video, not from writing to them. Contemporary Educational Psychology 2016; 44-45:95–106.

Jacob L, Lachner A, Scheiter K. Learning by explaining orally or in written form? Text complexity matters. Learning and Instruction 2020; 68:101344.

Jerusalem M, Schwarzer R. Allgemeine Selbstwirksamkeitserwartung. In: Schwarzer R, Jerusalem M, editors. Skalen zur Erfassung von Lehrer- und Schülermerkmalen: Dokumentation der psychometrischen Verfahren im Rahmen der Wissenschaftlichen Begleitung des Modellversuchs Selbstwirksame Schulen. Berlin; 1999. p. 13–4.

Jerusalem M, Schwarzer R. Allgemeine Selbstwirksamkeitserwartung. In: Schwarzer R, Jerusalem M, editors. Skalen zur Erfassung von Lehrer- und Schülermerkmalen: Dokumentation der psychometrischen Verfahren im Rahmen der Wissenschaftlichen Begleitung des Modellversuchs Selbstwirksame Schulen. Berlin; 1999. p. 13–4.

Kiebel, S, J, Holems, A, P. The general linear model. In: Penny, W, Friston, K, Ashburner, J, T. Statistical Parametric mapping: The analysis of functional brain images. Elsevier professional. p. 101-125.

Kim Y. Digital peers to help children's text comprehension and perceptions. Educational Technology & Society 2013; 16(4):59–70.

Kunter M, Baumert J, Blum W, Klusmann U, Krauss S, Neubrand M, editors. Cognitive activation in the mathematics classroom and professional competence of teachers: Results from the COACTIV project. New York: Springer; 2013. (Mathematics teacher education; vol 8). 

Lachner A, Jacob L, Hoogerheide V. Learning by writing explanations: Is explaining to a fictitious student more effective than self-explaining? Learning and Instruction 2021.

Lazarides R, Gaspard H, Dicke A-L. Dynamics of classroom motivation: Teacher enthusiasm and the development of math interest and teacher support. Learning and Instruction 2019; 60:126–37.

Oh CS, Bailenson JN, Welch GF. A systematic review of social presence: Definition, antecedents, and implications. Front Robot AI 2018; 5:114.

Paas F. Training strategies for attaining transfer of problem-solving skill in statistics: A cognitive-load approach. Journal of Educational Psychology 1992; 84(4):429–34.

Ranganathan P, Pramesh CS, Buyse M. Common pitfalls in statistical analysis: The perils of multiple testing. Perspect Clin Res 2016; 7(2):106–7.

Savary J, Kleiman T, Hassin RR, Dhar R. Positive consequences of conflict on decision making: When a conflict mindset facilitates choice. J Exp Psychol Gen 2015; 144(1):1–6.

Seidel T, Stürmer K, Blomberg G, Kobarg M, Schwindt K. Teacher learning from analysis of videotaped classroom situations: Does it make a difference whether teachers observe their own teaching or that of others? Teaching and Teacher Education 2011; 27(2):259–67.

Sindoni MG. Spoken and written discourse in online interactions: A multimodal approach. New York, NY: Routledge; 2013.

Short J, Williams E, Christie B. The social psychology of telecommunications. London: Wiley; 1976.

Streiner DL. Best (but oft-forgotten) practices: The multiple problems of multiplicity-whether and how to correct for many statistical tests. Am J Clin Nutr 2015; 102(4):721–8.

Wang J, Antonenko PD. Instructor presence in instructional video: Effects on visual attention, recall, and perceived learning. Computers in Human Behavior 2017; 71:79–89.

Weinhuber M, Lachner A, Leuders T, Nückles M. Mathematics is practice or argumentation: Mindset priming impacts principle- and procedure-orientation of teachers' explanations. J Exp Psychol Appl 2019; 25(4):618–46.

Widmer, M, Ziegler, N, Held, J, Luft, A, Lutz, K. Rewarding feedback promotes motor skill consolidation via striatal activity. Progress in Brain Research 2016; 229: 303-323.

---

## [Decision Letter · Decision Letter 1]

31 Mar 2021

PONE-D-20-39683R1

Does increasing social presence enhance the effectiveness of writing explanations?

PLOS ONE

Dear Dr. Jacob,

Thank you for submitting your manuscript to PLOS ONE. After careful consideration, we feel that it has merit but does not fully meet PLOS ONE’s publication criteria as it currently stands. Therefore, we invite you to submit a revised version of the manuscript that addresses the points raised during the review process.

I see that the previous suggestions made by the reviewers were followed. However, there are still some minor revisions suggested by Reviewer #2 that should be addressed before I recommend this manuscript for publication. If these minor revisions are accepted and addressed, I might grant acceptance without sending it to further external peer review.

We look forward to receiving your revised manuscript.

Kind regards,

Juan Cristobal Castro-Alonso, Ph.D.

Academic Editor

PLOS ONE

Journal Requirements:

Reviewers' comments:

Reviewer's Responses to Questions

**Comments to the Author**

1. If the authors have adequately addressed your comments raised in a previous round of review and you feel that this manuscript is now acceptable for publication, you may indicate that here to bypass the “Comments to the Author” section, enter your conflict of interest statement in the “Confidential to Editor” section, and submit your "Accept" recommendation.

Reviewer #1: All comments have been addressed

Reviewer #2: (No Response)

2. Is the manuscript technically sound, and do the data support the conclusions?

Reviewer #1: Yes

Reviewer #2: Yes

3. Has the statistical analysis been performed appropriately and rigorously? 

Reviewer #1: Yes

Reviewer #2: Yes

4. Have the authors made all data underlying the findings in their manuscript fully available?

Reviewer #1: Yes

Reviewer #2: Yes

5. Is the manuscript presented in an intelligible fashion and written in standard English?

Reviewer #1: Yes

Reviewer #2: Yes

6. Review Comments to the Author

Reviewer #1: Dear authors,

thank you for this thorough and overall impressive revision. I think the additions you made in this revision add to the overall quality of the manuscript and I feel that my comments have been adequately addressed. In total, I think this is important and rigorous work and I'm happy to recommend acceptance.

Reviewer #2: The authors addressed all concerns raised by the reviewers in their revision. I have only some minor comments.

Page 8

The results of previous studies are described in too much detail. I think that details of the types of statistical analyses applied are not relevant or even not easily comprehensible (e.g., contrast analyses, serial mediation analyses).

Page 12

The reliability of the knowledge tests is relatively low. Please explain why. In this context, please explain why you used McDonald’s reliability index.

Page 14

The authors provide a rho coefficient to indicate reliability of the tests. Please explain what this coefficient means.

Discussion

The authors found that all students overestimated their performance in the knowledge test. However, the authors do not explain why students did so. Please explain.

I think another limitation of the authors’ study is that students who were not familiar with the topic to be explained served as participants. Given that explaining is usually a task that is not done routinely (Rozenblit & Keil, 2002), I wondered whether these students were generally interested in explaining an unfamiliar topic. Maybe, this fact might also explain the null results for learning.

Rozenblit, L., & Keil, F. (2002). The misunderstood limits of folk science: An illusion of explanatory depth. Cognitive Science, 26(5), 521–562. https://doi.org/10.1207/s15516709cog2605_1

7. PLOS authors have the option to publish the peer review history of their article (what does this mean?). If published, this will include your full peer review and any attached files.

Reviewer #1: **Yes: **Joshua Weidlich

Reviewer #2: No

---

## [Author Response · Author response to Decision Letter 1]

1 Apr 2021

Dear Dr. Castro-Alonso, 

Dear reviewers,

Thank you very much for your editorial efforts and the opportunity to resubmit our manuscript. Hereby, we would like to submit our revision in which we addressed all the issues raised by the reviewers. We carefully revised our manuscript accordingly and responded to each comment with utmost care. For readability, the reviewers’ comments are printed in bold and responses in regular font. Furthermore, we additionally highlighted all changes in the manuscript in quotation marks in the rebuttal letter.

We are convinced that the reviewers’ comments helped us improve our work and hope that we revised the manuscript as expected.

Yours sincerely,

The Authors

Reviewer #1

Dear authors,

thank you for this thorough and overall impressive revision. I think the additions you made in this revision add to the overall quality of the manuscript and I feel that my comments have been adequately addressed. In total, I think this is important and rigorous work and I'm happy to recommend acceptance.

Thank you very much for your appreciative words and your work which helped us improving our manuscript. 

Reviewer #2

The authors addressed all concerns raised by the reviewers in their revision. I have only some minor comments.

We addressed all comments with utmost care.

Page 8

The results of previous studies are described in too much detail. I think that details of the types of statistical analyses applied are not relevant or even not easily comprehensible (e.g., contrast analyses, serial mediation analyses).

We revised the corresponding paragraph accordingly by avoiding specific analyses and reformulated sentences to facilitate the reading process (p. 7-8).

“Results revealed an interaction effect between learning activity and text difficulty (d = 0.31). The effect of explaining was only significant in the high-difficult condition, but not in the low-difficult condition. Thus, the explaining effect only held true when students learned from difficult but not from less difficult material. More interestingly, students who explained the contents to a fictitious peer orally outperformed students who wrote an explanation. Again, this effect was only significant when the learning material was difficult, but not when it was less complex. Additionally, perceived social presence and the richness of explanations mediated the explanatory effect: Students who explained orally perceived a stronger presence of the fictitious peer (measured by the number of personal references) compared to students who wrote an explanation. Higher levels of social presence, in turn, was associated with richer explanations (measured by the number of mentioned concepts) which resulted in better learning outcomes, at least for the difficult text.”

Page 12

The reliability of the knowledge tests is relatively low. Please explain why. In this context, please explain why you used McDonald’s reliability index.

We used the knowledge tests for measuring prior-knowledge and comprehension referring to different non-necessarily related sub-topics to cover students’ comprehensive understanding. In other words, we measured a broad knowledge base of the topic “immunology” by asking different and independent questions regarding different subtopics of immunology. This pattern of questions will necessarily result in low internal consistency (Tavakol & Dennick, 2011), as there is a trade-off between comprehensiveness and consistency. Therefore, we would argue from a theoretical perspective that low consistencies say little about the quality of the measurement (see also Scheiter et al., 2015; Stadler et al., 2021). Nevertheless, we discussed the low reliability values in the limitation section on pages 28-29 in the revised manuscript.

“Another limitation refers to the relatively low homogeneity indices. However, we used validated tests which were implemented in prior studies. Additionally, we want to note that we measured a broad understanding of the topic by asking different and independent questions with a restricted set of 5 to 6 items per test. This decision likely resulted in a trade-off regarding the internal consistency of the knowledge measures, as they likely covered diverse sub-components of students’ comprehension [1]. Further research with advanced statistical procedures is needed which explicitly takes the multidimensionality into account, for instance, by using bifactor-(S-1) measurement models which, however, require larger sample sizes [2].”

Regarding the homogeneity indices, we want to note that using Cronbach’s alpha is discussed as rather problematic for determining the reliability of knowledge tests. For instance, Cronbach’s alpha is commonly regarded as appropriate measure for the internal consistency of one single construct (Dunn et al, 2014) but not of multidimensional construct as in the present study. McDonald’s Omega, in contrast, is an adequate measure for reliability for multidimensional constructs (Hayes & Coutts, 2020). Besides the revised discussion paragraph, we additionally highlighted the multidimensionality of our constructs on page 12 in the revised manuscript:

“We used the prior knowledge test from Golke and Wittwer [3], which contained five questions with an open-ended answer format and represented a multidimensional construct, measuring different subcomponents of immunology (e.g., “Why can viruses be dangerous for humans?”; McDonald’s ωt = .60). […] We measured students’ text-based knowledge and inference knowledge with two different posttests from Golke and Wittwer [3]. The tests consisted of different questions than the prior knowledge test and represented multidimensional constructs.”

Dunn, T. J., Baguley, T., & Brunsden, V. (2014). From alpha to omega: A practical solution to the pervasive problem of internal consistency estimation. British Journal of Psychology, 105, 399-412.

Hayes, F. A., & Coutts, J. J. (2020). Use omega rather than cronbach’s alpha for estimating reliability. But…. Communication Methods and Measures, 14(1), 1-24.

Scheiter, K., Schüler, A., Gerjets, P., Huk, T., & Hesse, F. W. (2015). Extending multimedia research: How do prerequisite knowledge and reading comprehension affect learning from text and pictures. Computer in Human Behavior, 31, 73-84.

Stadler, M., Sailer, M., & Fischer, F. (2021). Knowledge as a formative construct: A good alpha is not always better. New Ideas in Psychology, 60, 100832.

Tavakol, M., & Dennick, R. (2011). Making sense of Cronbach's alpha. International Journal of Medical Education, 2, 53-55.

Page 14

The authors provide a rho coefficient to indicate reliability of the tests. Please explain what this coefficient means.

We calculated Spearman Brown’s rho for enjoyment and interest as both scales contain only two items and rho presents a correlative measure for reliability. However, in terms of cohesion, we now calculated McDonalds Omega (p. 14).

“We measured students’ enjoyment during the learning activity by using two self-generated items (e.g., “I enjoyed doing the task”). The students rated their enjoyment on a 4-point Likert scale from 1 “not at all” to 4 “absolutely”. Reliability was good (McDonald’s ωt = .96). […] We measured students’ enjoyment during the learning activity by using two self-generated items (e.g., “I enjoyed doing the task”). The students rated their interest on a 4-point Likert scale from 1 “not at all” to 4 “absolutely”. Again, reliability was good (McDonald’s ωt = .81).”

Discussion

The authors found that all students overestimated their performance in the knowledge test. However, the authors do not explain why students did so. Please explain.

Prior research demonstrated that students generally overestimate their current understanding (Eitel, 2016; Mayer et al., 2007; Prinz et al., 2020). Thus, we did not attribute this finding to our study but rather to students’ tendency of overestimating their comprehension. We discussed the finding in our discussion section on page 25 in the revised manuscript.

“Additionally, students showed comparable monitoring accuracy ratings across conditions. Interestingly, all students overestimated their current understanding. This, however, is in line with prior research that highlighted that students generally tend to overestimate their current comprehension [4–6]. Nevertheless, we need to reject our hypotheses regarding effects on students’ comprehension and their monitoring accuracy.”

I think another limitation of the authors’ study is that students who were not familiar with the topic to be explained served as participants. Given that explaining is usually a task that is not done routinely (Rozenblit & Keil, 2002), I wondered whether these students were generally interested in explaining an unfamiliar topic. Maybe, this fact might also explain the null results for learning.

Rozenblit, L., & Keil, F. (2002). The misunderstood limits of folk science: An illusion of explanatory depth. Cognitive Science, 26(5), 521–562. https://doi.org/10.1207/s15516709cog2605_1

Explaining new and unfamiliar contents is a common learning activity in research on generative learning activities (Brod, 2020; Fiorella & Mayer, 2016). Thus, our approach did not differ from prior research (see Fiorella & Mayer, 2013, 2014; Fukaya, 2013; Hoogerheide et al., 2016, 2019; Lachner et al., 2020, 2021). Nevertheless, our null findings might be due to low levels of prior knowledge. However, prior research revealed that generating an explanation is particularly beneficial for students with low prior knowledge (see Hoogerheide et al., 2019). We discussed students’ low prior knowledge in relation to the null findings in the discussion section on page 25 in the revised manuscript.

“The null findings regarding students’ comprehension might be attributed to low levels of prior knowledge. Little prior knowledge limits students in learning new contents adequately [7, 8]. However, prior research revealed that generative learning activities, such as explaining, are in particular beneficial for low prior knowledge students [9].”

We discussed the challenge of generating an explanation in the discussion section on page 27 in the revised manuscript.

“Nevertheless, against the background of the relatively low quality of students’ explanations, as a theoretical consequence of our study, we suggest that future research should focus on how students could be supported to generate high-quality explanations. Given that students are generally not familiar with explaining challenging contents [10], they might depend on further support to be able to apply this learning activity successfully. As a first attempt, Lachner and Neuburg [11] investigated the role of formative feedback during explaining (in written form). They found that formative feedback helped students generate more cohesive explanations, which finally contributed to their comprehension. Additionally, the self-explaining literature focusses on two main instructional approaches [12], supporting self-explaining directly by means of pre-trainings [13] or indirectly by the use of prompts [14, 15], which should work as strategy activators to enact deep-level explaining strategies. Whether and under which conditions such direct and indirect support procedures also hold true for generating explanations to fictitious others has to be investigated by further research.”

Brod, G. (2020). Generative learning: Which strategies for what age? Educational Psychology Review. Advance online publication. https://doi.org/10.1007/s10648-020-09571-9

Fiorella, L., & Mayer, R. E. (2013). The relative benefits of learning by teaching and teaching expectancy. Contemporary Educational Psychology, 38(4), 281–288. https://doi.org/10.1016/j.cedpsych.2013.06.001

Fiorella, L., & Mayer, R. E. (2014). Role of expectations and explanations in learning by teaching. Contemporary Educational Psychology, 39(2), 75–85. https://doi.org/10.1016/j.cedpsych.2014.01.001

Fiorella, L., & Mayer, R. E. (2016). Eight ways to promote generative learning. Educational Psychology Review, 28(4), 717–741. https://doi.org/10.1007/s10648-015-9348-9

Fukaya, T. (2013). Explanation generation, not explanation expectancy, improves metacomprehension accuracy. Metacognition Learning, 8(1), 1–18. https://doi.org/10.1007/s11409-012-9093-0

Hoogerheide, V., Deijkers, L., Loyens, S. M., Heijltjes, A., & van Gog, T. (2016). Gaining from explaining: Learning improves from explaining to fictitious others on video, not from writing to them. Contemporary Educational Psychology, 44-45, 95–106. https://doi.org/10.1016/j.cedpsych.2016.02.005

Hoogerheide, V., Renkl, A., Fiorella, L., Paas, F., & van Gog, T. (2019). Enhancing example-based learning: Teaching on video increases arousal and improves problem-solving performance. Journal of Educational Psychology, 111(1), 45–56. https://doi.org/10.1037/edu0000272

Lachner, A., Backfisch, I., Hoogerheide, V., van Gog, T., & Renkl, A. (2020). Timing matters! Explaining between study phases enhances students’ learning. Journal of Educational Psychology, 112(4), 841–853. https://doi.org/10.1037/edu0000396

Lachner, A., Jacob, L., & Hoogerheide, V. (2021). Learning by writing explanations: Is explaining to a fictitious student more effective than self-explaining? Learning and Instruction.

 

References

1. Dunn TJ, Baguley T, Brunsden V. From alpha to omega: A practical solution to the pervasive problem of internal consistency estimation. British Journal of Psychology 2014; 105(3):399–412.

2. Eid M, Geiser C, Koch T, Heene M. Anomalous results in G-factor models: Explanations and alternatives. Psychol Methods 2017; 22(3):541–62.

3. Golke S, Wittwer J. High-performing readers underestimate their text comprehension: Artifact or psychological reality? Proceedings of the 39th Annual Conference of the Cognitive Science 2018.

4. Eitel A. How repeated studying and testing affects multimedia learning: Evidence for adaptation to task demands. Learning and Instruction 2016; 41:70–84.

5. Mayer RE, Stull AT, Campbell J, Almeroth K, Bimber B, Chun D et al. Overestimation bias in self-reported SAT scores. Educational Psychology Review 2007; 19(4):443–54.

6. Prinz A, Golke S, Wittwer J. To What Extent Do Situation-Model-Approach Interventions Improve Relative Metacomprehension Accuracy? Meta-Analytic Insights. Educ Psychol Rev 2020; 32(4):917–49.

7. Brod G. Generative learning: Which strategies for what age? Educational Psychology Review 2020.

8. Renkl A. Lehren und Lernen. In: Tippelt R, Schmidt-Hertha B, editors. Handbuch Bildungsforschung. Wiesbaden: Verlag für Sozialwissenschaften; 2009. p. 737–51.

9. Hoogerheide V, Renkl A, Fiorella L, Paas F, van Gog T. Enhancing example-based learning: Teaching on video increases arousal and improves problem-solving performance. Journal of Educational Psychology 2019; 111(1):45–56.

10. Rozenblit L, Keil F. The misunderstood limits of folk science: An illusion of explanatory depth. Cogn Sci 2002; 26(5):521–62.

11. Lachner A, Neuburg C. Learning by writing explanations: Computer-based feedback about the explanatory cohesion enhances students’ transfer. Instructional Science 2019; 47(1):19–37.

12. Fonseca BA, Chi MTH. Instruction based on self-explanation. In: Mayer RE, Alexander PA, editors. Handbook of research on learning and instruction. New York: Routledge; 2011 (Educational psychology handbook series).

13. McNamara DS. Self-explanation and reading strategy training (SERT) improves low-knowledge students’ science course performance. Discourse Processes 2017; 54(7):479–92.

14. Chi MTH, Leeuw N de, Chiu M-H, LaVancher C. Eliciting self-explanations improves understanding. Cogn Sci 1994; 18(3):439–77.

15. Schworm S, Renkl A. Learning argumentation skills through the use of prompts for self-explaining examples. Journal of Educational Psychology 2007; 99(2):285–96.

---

## [Editor Report · Decision Letter 2]

7 Apr 2021

Does increasing social presence enhance the effectiveness of writing explanations?

PONE-D-20-39683R2

Dear Dr. Jacob,

We’re pleased to inform you that your manuscript has been judged scientifically suitable for publication and will be formally accepted for publication once it meets all outstanding technical requirements.

Kind regards,

Juan Cristobal Castro-Alonso, Ph.D.

Academic Editor

PLOS ONE

---

## [Editor Report · Acceptance letter]

14 Apr 2021

PONE-D-20-39683R2 

Does increasing social presence enhance the effectiveness of writing explanations? 

Dear Dr. Jacob:

I'm pleased to inform you that your manuscript has been deemed suitable for publication in PLOS ONE. Congratulations! Your manuscript is now with our production department. 

Kind regards, 

on behalf of

Dr. Juan Cristobal Castro-Alonso 

Academic Editor

PLOS ONE